# A Two-Level WiFi Fingerprint-Based Indoor Localization Method for Dangerous Area Monitoring

**DOI:** 10.3390/s19194243

**Published:** 2019-09-29

**Authors:** Fei Li, Min Liu, Yue Zhang, Weiming Shen

**Affiliations:** 1Department of Computer Science, Zhejiang University City College, Hangzhou 310015, China; 2College of Electronics and Information Engineering, Tongji University, Shanghai 201804, China; lmin@tongji.edu.cn (M.L.); yuezhang@tongji.edu.cn (Y.Z.); 3State Key Laboratory of Digital Manufacturing Equipment and Technology, School of Mechanical Science and Engineering, Huazhong University of Science and Technology, Wuhan 430074, China; wshen@ieee.org

**Keywords:** disaster management, disaster relief, indoor fingerprint localization, affinity propagation clustering (APC), support vector regression (SVR), particle swarm optimization (PSO)

## Abstract

Localization technologies play an important role in disaster management and emergence response. In areas where the environment does not change much after an accident or in the case of dangerous areas monitoring, indoor fingerprint-based localization can be used. In such scenarios, a positioning system needs to have both a high accuracy and a rapid response. However, these two requirements are usually conflicting since a fingerprint-based indoor localization system with high accuracy usually has complex algorithms and needs to process a large amount of data, and therefore has a slow response. This problem becomes even worse when both the size of monitoring area and the number of reference nodes increase. To address this challenging problem, this paper proposes a two-level positioning algorithm in order to improve both the accuracy and the response time. In the off-line stage, a fingerprint database is divided into several sub databases by using an affinity propagation clustering (APC) algorithm based on Shepard similarity. The online stage has two steps: (1) a coarse positioning algorithm is adopted to find the most similar sub database by matching the cluster center with the fingerprint of the node tested, which will narrow the search space and consequently save time; (2) in the sub database area, a support vector regression (SVR) algorithm with its parameters being optimized by particle swarm optimization (PSO) is used for fine positioning, thus improving the online positioning accuracy. Both experiment results and actual implementations proved that the proposed two-level localization method is more suitable than other methods in term of algorithm complexity, storage requirements and localization accuracy in dangerous area monitoring.

## 1. Introduction

Indoor localization plays an important role in public safety and disaster management and relief [1]. Location data can help managers warn people who make unauthorized entry into areas to enhance public safety. Such situations include the monitoring of dangerous military areas or indoor inflammable stacking areas. Another example is the fire emergence response: when a fire incident occurs in a building, management personnel can locate both the disaster site and people quickly [2]. During disaster relief operations, location information can assist rescuers to quickly acquire tools, get closer to those to be rescued and evacuate them [3,4].

Common indoor positioning technologies include range-based and non-range-based [5,6]. The former is realized by measuring the physical relationship between the measurable parameters and the position of the signal arrival time and signal strength. The latter includes two phases: offline and online. During the offline phase, different signal strengths are received for different locations in the room, which is used to generate a wireless signal fingerprint radio map. During the online phase, a real-time signal fingerprint received from the actual location is matched with the fingerprint database to obtain the fingerprint with the highest similarity, thereby estimating the target position. At present, wireless local area network (WLAN) -based indoor fingerprint positioning is the main technology, largely due to the WiFi availability and its low cost.

The indoor positioning methods available for different disaster sites are different. For instance, in the event of a fire in a building, the environment changes quickly: such a situation is not suitable for received signal strength indication (RSSI) ranging or RSSI fingerprinting. Instead, pedestrian dead reckoning (PDR) positioning [2] or other wearable sensing networks [7] are used. However, indoor fingerprint positioning can achieve good results where the environment does not change quickly, such as monitoring dangerous areas for unauthorized entry or for chemical poison or toxin gas leakage.

No matter which indoor positioning technology is adopted in disaster management, its primary requirements are accuracy and real-time response [1,2,3,4,8]. Accurate positioning can avoid false positives and false negatives when, at a disaster site, response personnel may have less than one minute for rescue or evacuation [4]. However, the accuracy and real-time performance of indoor positioning is usually conflicting. Generally speaking, if positioning accuracy needs to be improved, more complex algorithms or more reference data are required [2,9]. Conversely, if a simpler algorithm is adopted, the estimated location may not be accurate enough. Thus, choosing a good localization algorithm is important but not easy in disaster management.

Taking, as an example, an indoor fingerprint localization algorithm applied to hazardous area monitoring and disaster relief systems in a chemical plant, this paper proposed a two-level positioning algorithm based on improved clustering and particle swarm optimization (PSO)-support vector regression (SVR) to improve fingerprint matching efficiency. It specifically investigated two issues. Firstly, in the offline phase, the similarity among the fingerprints was measured by the Shepard similarity which stems from the Euclidean distance and the fingerprint database was divided into sub-fingerprint databases by the affinity propagation clustering (APC) algorithm. The localization model based on PSO-SVR was then established for each fingerprint database, where the uncertainty of the parameter selection of the SVR regression model was optimized by the PSO algorithm. Secondly, in the online phase, the most similar sub-fingerprint database was firstly selected according to the similarity between the target point fingerprint and the sub-fingerprint cluster center fingerprint. Within this sub-area, the PSO-SVR was then used again to obtain the accurate location of the tested point. It should be noted that both improved accuracy and the time reduction in the online phase came at the expense of increased workload in the offline phase.

The rest of this paper is arranged as follows: Section 2 reviews the related literature; Section 3 introduces the proposed localization algorithm, including the overall framework of the proposed method, APC algorithm based on the Shepard similarity metric and accurate localization based on PSO-SVR; Section 4 reports the results of performance evaluation based on a real application; Section 5 provides conclusions and discusses future work.

## 2. Review of Related Literature 

A wireless fingerprint indoor positioning method usually has two phases: offline and online. The positioning performance of the online phase determines the overall performance of the system and its positioning efficiency. Commonly used matching algorithms include K-nearest neighbor (KNN) [10], kernel-based algorithm [11], support vector machine [12], artificial neural network (ANN) [13,14], and SVR [15].

In terms of performance, a localization system using the above algorithms usually cannot simultaneously meet the requirements of localization accuracy, algorithm complexity and storage reduction. For example, the KNN algorithm is simple but not precise enough, while ANN has high positioning accuracy but its algorithm is complex and needs much storage. Research shows that the localization accuracy of the non-parametric methods depends largely on the number of samples [16,17]. However, the “flat”, one-by-one matching method increases matching calculations and reduces the matching efficiency for fingerprints [5,18] and thus affects the overall efficiency of the system.

Since 2014, two-level localization algorithms have been widely adopted in the fields of disaster management [2,19] and other fields [9,20,21]. Research and experimentation [13,22,23] show that, according to the similarity between target-position fingerprints and offline-collected fingerprints, the number of fingerprints matched can be effectively reduced [24,25] by selectively matching some appropriate fingerprints, thus establishing a hierarchical search algorithm. This can simultaneously achieve considerable precision. To achieve this, the positioning database is divided into a certain number of sub-databases in the off-line stage to achieve database partition [26]. In the online stage, the matching range is narrowed by searching the sub-databases adjacent to the target point [27]. Therefore, there are typically two kinds of important algorithms: one is the clustering algorithm which is critical to sub-database partition in the offline stage, and the other is the classification algorithm of fine localization in the online stage.

Although there are many algorithms proposed and developed in the literature for clustering, such as K-Means, density-based spatial clustering of applications with noise (DBSCAN), the expectation maximization (EM) based on Gaussian mixture model (GMM), and APC, for algorithm simplicity and localization speed, the K-Means and APC algorithms are the most widely used in indoor fingerprint localization no matter whether in “flat” or hierarchical localization methods.

The K-Means is considered to be the simplest of all clustering algorithms, and was also used in a two-level localization algorithm [28], but the initial value of the K parameter of K-Means clustering is difficult to determine, and it cannot cluster samples with complex distribution, such as non-convex distributed samples. So, in some instances, it is not appropriate to use the K-Means for clustering.

An alternative solution is the APC algorithm, which can naturally form the number of clusters, hence avoiding the problem of some parameters presetting. Currently, some non-hierarchical localization algorithms [1,21,29,30] use APC as the clustering algorithm. In some hierarchical localization algorithms, APC is also used [9,22]. All the current APC algorithms use the Euclidean distance as the similarity metric, which stems from the most primitive algorithm [31]. However, according to the literature [9,32], RSSI does not have a linear relationship with the Euclidean distance, so it is unreasonable to measure the RSSI similarity with the Euclidean distance metric. This will be further discussed in Section 3.2.1. 

The second kind of important algorithms in hierarchical localization is classification. Commonly, weighted *k*-nearest neighbor (W*k*NN) [9,20], SVR [22] and ANN [33] are adopted. W*k*NN is the most popular used algorithm, but sometimes it cannot achieve good accuracy. On the contrary, ANN is the most accurate, but it is the most complex. SVR is an efficient classification algorithm due to its high generalization ability and excellent ability of non-linear mapping with the kernel function.

Compared with a “flat” localization algorithm, all the classification algorithms in hierarchical localization can save operation time because of their reduced search range. However, all these algorithms still face the same problem as the K-Means clustering algorithm: the parameters used in the algorithms must be preset in advance, such as the *k* parameter in W*k*NN and the insensitivity coefficient ɛ in SVR. If the parameters are not properly preset, accuracy will deteriorate.

## 3. Proposed Indoor Localization Method

### 3.1. Overall Framework of the Proposed Method

As mentioned above, the proposed localization algorithm was divided into two phases: offline and online (as shown in Figure 1). 

The offline phase included the following steps:
(1)RSSI data collection and pre-processing: collecting RSSI fingerprint data multiple times, and filter them to reduce fluctuations resulting from changing indoor temperature, humidity, multipath effect and other factors.(2)Data segmentation: the offline fingerprint data, clustered by the APC algorithm, is segmented into different fingerprint subsets according to the Shepard-similarity between the fingerprint data.(3)Localization model creation: in different fingerprint subsets, the PSO-SVR algorithm is used for training to obtain a localization model.


The online stage includes the following steps:
(1)RSSI data collection and pre-processing: RSSI signals are collected and pre-processed in a similar way as in the offline phase.(2)Coarse localization: according to the cluster center similarity metric, the similarity between the fingerprint of the target point and each cluster center is calculated and the fingerprint subset with the highest similarity is selected for further localization.(3)Fine localization: the fingerprint data of the target point is input into the positioning model and the coordinates of the target position are calculated, thereby completing the localization.


Please note that the focus of this paper was on localization algorithms. Due to the space limitation, four aspects of the problem and the proposed solution are briefly introduced as follows:
(1)Filtering during the data pre-processing: as described in [34], filtering is critical to the positioning accuracy. After having compared various methods including median filtering, moving average filtering, and Gaussian filtering, we believe that Gaussian filtering is better than the other two methods in that it has smaller variance and more smoothing results. Therefore, the Gaussian filtering has been chosen as the proposed data pre-processing method.(2)Access point (AP) selection: it is not always good to choose all APs, and whether an AP is selected as a fingerprint depends on two aspects: (1) whether its fingerprint vector is stable, since there are significant differences in RSSI values collected between online and offline stages, resulting in large positioning errors; (2) whether it has a high number of occurrences since this determines the total error resulted from the differences between default values and real values. An indicator is defined as follows:
(1)$$Indicator_{j}=\eta_{AP_{j}}\times \frac{1}{SD_{j}+\delta }$$
where $\eta_{AP_{j}}$ is the occurrence frequency of ${\mathrm{AP}}_{j}$; $SD_{j}$ is the standard deviation of RSSI of ${\mathrm{AP}}_{j}$; δ is a minimal positive number set to prevent the zero denominator of Formula (1). $SD_{j}$ is defined as following:
(2)$$SD_{j}=\sqrt{\frac{1}{n-1}\sum_{j=1}^{n}{\left(RSSI_{ij}-\bar{RSSI_{j}}\right)}^{2}}$$
where *RSSI_i_*_j_ is the ith sampling value of the ${\mathrm{AP}}_{j}$, and $\bar{RSSI_{j}}$ is the mean value of the sampling value of the ${\mathrm{AP}}_{j}$. After calculating indicators of all APs and sorting them, the largest APs were selected as features of the fingerprint vectors.(3)Matching in the coarse localization stage: there are two kinds of criteria for coarse positioning as reference states [9,21]: (1) cluster center similarity criterion, which calculates the similarity between real-time fingerprint and each cluster center according to the similarity definition, and select the cluster with the highest similarity as the cluster matching result; (2) the average fingerprint similarity criterion, which firstly calculates the average value of the every cluster fingerprint, and then calculates the similarity between the real-time fingerprint and the average value of each cluster according to the similarity definition, and selects the cluster with the highest similarity as the cluster matching result. For the sake of simplicity of the algorithm, this paper adopted the first criterion as the cluster matching criterion in the coarse localization stage.(4)Radio map building: the common method of establishing a fingerprint database is to manually collect RSSI fingerprint data at a fixed reference point (RP). Based on the results from [35], usually, the denser the RPs deployed in the target location area, the higher the localization accuracy. However, such a manual method leads to a complicated position system and a heavier workload in the offline phase. This paper adopted the Gaussian Process Regression (GPR) model to predict the fingerprint of some virtual RP points based on collection of limited real RP points, thus reducing the workload when building the radio map.


### 3.2. APC Algorithm Based on the Shepard Similarity Metric

#### 3.2.1. The Shepard Similarity Metric

The clustering process classified sample points according to their similarity. It required the similarity within the cluster to be as large as possible and the similarity between clusters to be as small as possible. Therefore, calculating the similarity of clustering objects was one of the most important tasks of the clustering algorithm. There are various similarity measurements: Minkowski distance, cosine similarity, Pearson similarity and Jaccard similarity [9,31]. A Minkowski distance can be denoted as:
(3)$$D^{p}(s_{i},s_{j})=(\sum_{l=1}^{L}{\left|s_{l,i}-s_{l,j}\right|}^{p}{)}^{\frac{1}{p}}p\geq 1$$
where $s_{i}$ and $s_{j}$ represent fingerprints of two points in *l* dimensions and, if *p* = 1, then that distance is the Manhattan distance and *p* = 2 is the Euclidean distance. The format of $s_{i}$ is defined as follows:
(4)$$s_{i}=(RSSI_{1},RSSI_{2},\cdots ,RSSI_{l})$$
where the fingerprint of the referent point *i* includes RSSIs received from *l* APs.

But the situation in indoor fingerprint location is a little different from the above. The power attenuation model of the wireless signal is as follows.
(5)$$PL(d)=PL(d_{0})-10n\mathrm{lg}(\frac{d}{d_{0}})+\varsigma$$
where *PL*(*d*) is received RSSI at the distance *d* from the anchor point; *PL*(*d*_0_) represents the received RSSI when the distance is *d*_0_ from the anchor point; *ζ* is the random variable compensating for fluctuations and *n* is the path loss coefficient. As can be seen from the above formula, RSSI and distance *d* satisfy the property of a logarithmic relationship rather than a linear relationship. Therefore in indoor fingerprint localization, the method used Formula (3) is not suitable for RSSI signal similarity measurement. In order to punish the RSSI role of distant node signals, weak signals and noise signals in localization, the node signal similarity is defined as the exponential form of Euclidean distance—the Shepard similarity:
(6)$$S_{Shepard}(s_{i},s_{j})=e^{-D^{2}(s_{i,},s_{j})}$$


The similarity of distant nodes can thus be rapidly reduced, eliminating the superposition of distant signals in the estimation of the node position.

#### 3.2.2. The APC Algorithm Based on Shepard Similarity

The APC algorithm is a very effective clustering algorithm. It differs from other traditional clustering algorithms, where the required number of categories and clustering centers are usually set at the beginning of the iteration of the clustering algorithm. In the APC algorithm, each data object has the same chance to become the clustering center, thus eliminating dependence on initial conditions.

In the APC clustering algorithm, two types of information are exchanged between data points: *i* and *j*: responsibility and availability. These two pieces of information are represented by the matrices *R* (*i*, *j*) and *A* (*i*, *j*), respectively. *R* (*i*, *j*) quantifies the confidence level at which the data points *j* are selected as the cluster center by the data points *i*; *A* (*i*, *j*) quantifies the appropriateness of the data points *i* to select the data points *j* as their cluster centers. In the process of iterative updating, the cluster centers and the corresponding clustering center of each data point are determined by *R* (*i*, *j*) and *A* (*i*, *j*) together.

In localization, responsibility and availability are denoted as $R(s_{i},s_{j})$ and $A(s_{i},s_{j})$, where $s_{i}$ has the fingerprint format as Formula (4) shows, and the calculation formulas of $R(s_{i},s_{j})$ and $A(s_{i},s_{j})$ are as follows:
(7)$$R(s_{i},s_{j})=S_{Shepard}(s_{i},s_{j})-\underset{n}{\max}[A(s_{i},s_{n})+S_{Shepard}(s_{i},s_{n})]$$
(8)$$A(s_{i},s_{j})=\min\{0,R(s_{j},s_{j})+\sum_{n}\max\{0,R(s_{n},s_{j})\}\}$$
where $S_{Shepard}(s_{i},s_{j})$ represents the Shepard similarity. ∀*i*, *j*, *n* ∈ {1,2,…,*N*}, and *i* ≠ *j*, *n* ≠ *j* in Equation (7), and *n* ≠ *i*, *j* in Equation (8), and *N* is the total number of RP.

$R(s_{i},s_{j})$ and $A(s_{i},s_{j})$ are updated as follows during the iteration:
(9)$$R^{new}(s_{i},s_{j})=\lambda \times R^{old}(s_{i},s_{j})+(1-\lambda )\times R(s_{i},s_{j})$$
(10)$$A^{new}(s_{i},s_{j})=\lambda \times A^{old}(s_{i},s_{j})+(1-\lambda )\times A(s_{i},s_{j})$$
where *λ* is a damping coefficient to prevent numerical oscillation, determining the robustness of the iterative process. When two or more data points can be selected as the clustering center of the same class, the clustering center of this class may oscillate between these data points, leading to the failure of the algorithm to converge. The value is generally set in (0.5, 1).

The specific process of the APC is presented as Algorithm 1:

**Algorithm 1**. The process of the affinity propagation clustering (APC) algorithm.
**# Input:**
 fingerprint data: *V* = [*S*_1_, *S*_2_, …, *S_n_*]^*T*^;
**# Output:**
 Cluster: *V* = {*C*_1_, *C*_2_, …, *C_k_*};
**# Initialization:**
 preference *ρ*; damping coefficient *λ*; Shepard similarity matrix **S** according to Formula (6); Availability matrix *A* = [0] n × n; Responsibility matrix *R* = [0] n × n;**for***k* = 1:num_iteration **if** stop condition is not satisfied  calculating the *R* (**s**_*i*_, **s**_*j*_) and *A* (**s**_*i*_, **s**_*j*_) according to Formula (7), (8);  updating the *R* (**s**_*i*_, **s**_*j*_) and *A* (**s**_*i*_, **s**_*j*_) according to Formula (9), (10); **end**
**end**

**# Exemplar identification:**
**for***k* = 1: *N* **if**
*R* (**s**_*i*_, **s**_*j*_) + *A* (**s**_*i*_, **s**_*j*_) > 0  Classifying the *RP_k_* as cluster center; **end**
**end**

**# Cluster assignments:**
Classifying other RP to corresponding cluster according to similarity;

The APC algorithm includes following steps:
(1)Initialization: Formula (6) is used to measure the similarity and calculate the similarity matrix S, and the damping coefficient *λ* is set to 0.9. The responsibility and availability matrices are initialized to zero matrix. The last parameter preference is defined thus:
(11)$$\rho =\gamma \times median\{sim_{Shepard}(s_{i},s_{j})\},\forall i,j\in \{1,2,\ldots ,N\},i\neq j$$
where *ρ* indicates preference, meaning the possibility of RP_*i*_ becoming a cluster center. Since all data points can be used as potential clustering centers in the initialization, data points have the same preference values at the beginning, which is usually a ratio of the median or minimum value of a similar matrix. The preference determines the number of clusters: the larger the preference, the more clusters will be generated. The coefficient γ is used to control the value of preference.(2)Iteration process: including the responsibility matrix and availability matrix update and iteration stop condition judgment.(3)Cluster center judgment and classification: determining whether it is a cluster center according to $A(s_{i},s_{j})+R(s_{i},s_{j})>0$, and classifying others’ RP according to similarity.


### 3.3. Accurate Localization Based on PSO-SVR

#### 3.3.1. The SVR Model and its Main Parameters

Due to the complexity of the indoor environment and the interference from noises such as people and multipath effects, the mapping between the high-dimensional signal strength space and the physical position is highly nonlinear in a localization algorithm based on fingerprints. The SVR algorithm was selected as the localization model in the fine positioning stage, performing well to solve nonlinear and high dimension problems.

The basic idea of SVR is to create a hyperplane, so that the distance from each point to the hyperplane is the shortest. For a given sample set $\left\{(x_{i},y_{i}),i=1,2,\cdots ,m\right\}$, where $x_{i}\in R^{n}$ is the n dimensions input vector and $y_{i}\in R$ is the corresponding output, the goal of SVR is to explore the mapping relationship that correctly reflects the existing relationship between input and output:
(12)$$f(x_{i})=\omega^{T}\Phi (x_{i})+b$$
where *f* (x) is the predictive output expression, *ɷ* is the adjustable weight vector and *b* denotes bias. $\Phi (x_{i}):R^{n}\to R^{d}$ is the nonlinear mapping of the input data in the sample set x_*i*_ from the original space $R^{n}$ to the high-dimensional feature space $R^{d}$.

In the SVR, the insensitive coefficient *ε* is used as the loss function, representing the tolerance of the error between the predicted value and the true value. An error “band” must be set up in order to contain all samples; it must be as narrow as possible. The middle line in the band is the regression line. The insensitive coefficient *ε* is shown in Figure 2.

Parameter *ε* plays an important role in performance: it directly determines the complexity and generalizability of the model, as well the number of support vectors used to construct it. In order to avoid over-fitting, parameter *ε* must be specially selected: the smaller the value, the more robust is the model, although there may be zero tolerance for training error. On the other hand, the bigger the value, the higher the tolerance to training error. To achieve a balance in this regard, it is necessary to use a set of relaxation variables $\xi ,\xi^{*}$ for the sample points outside the “band”—that is, which predictive error is greater than the parameter *ε*, so that the model can tolerate noises or outliers (Figure 2).

In order to obtain the mapping relationship between physical position and fingerprint vector, the kernel function is introduced to map the original fingerprint data from original space into a higher dimension space, making it linearly separable in this new space. The mathematical expression of the model is then:
(13)$$\begin{array}{l} \min\frac{1}{2}\omega^{T}\cdot \omega +C\sum_{k=1}^{n}\left(\xi_{k}+\xi_{k}{}^{*}\right)\\ s.t.\left\{ \begin{array}{l} y_{k}-(\omega^{T}\Phi (s_{k})+b)\leq \varepsilon +\xi_{k}\\ (\omega^{T}\Phi (s_{k})+b)-y_{k}\leq \varepsilon +\xi_{k}{}^{*}\\ \xi_{k},\xi_{k}{}^{*}\geq 0,i=1,\cdots ,N \end{array}\right. \end{array}$$
where *C* is the penalty factor, a constant that has been pre-set and *C* > 0; it determines the punishment degree of samples whose predictive error is greater than parameter *ε*. This is a convex quadratic optimization problem which can be solved by constructing a Lagrange function:
(14)$$\begin{array}{l} L(\omega ,b,\xi_{k},\xi_{k}^{*})=\frac{1}{2}\omega^{T}\cdot \omega +C\sum_{k=1}^{n}(\xi_{k}+\xi_{k}^{*})\\ -\sum_{k=1}^{n}\alpha_{k}(\varepsilon +\xi_{k}-y_{k}+\omega \cdot \Phi (s_{k})+b)\\ -\sum_{k=1}^{n}\alpha_{k}^{*}(\varepsilon +\xi_{k}^{*}+y_{k}-\omega \cdot \Phi (s_{k})-b)\\ -\sum_{k=1}^{n}\left(\lambda_{k}\cdot \xi_{k}+\lambda_{k}\cdot \xi_{k}^{*}\right) \end{array}$$
where, $\alpha_{k},\alpha_{k}^{*}(k=1,2,\ldots ,n)$ is the Lagrange multiplier. This is also a convex quadratic optimization problem. According to Karush-Kuhn-Tucker (KKT) conditions, we can derive:
(15)$$\left\{ \begin{array}{l} \frac{\partial L}{\partial \omega }=0\Rightarrow \omega =\sum_{k=1}^{n}\left(\alpha_{k}-\alpha_{k}^{*})\cdot \Phi (s_{k}\right)\\ \frac{\partial L}{\partial b}=0\Rightarrow \sum_{k=1}^{n}(\alpha_{k}-\alpha_{k}^{*})=0\\ \frac{\partial L}{\partial \xi }=0\Rightarrow \alpha_{k}=C\xi_{k}\\ \frac{\partial L}{\partial \xi^{*}}=0\Rightarrow \alpha_{k}^{*}=C\xi_{k}^{*} \end{array}\right.$$


After replacing Formula (14) with the value of each parameter variable in Formula (15), this dual optimization problem can be obtained:
(16)$$\begin{array}{l} \min\{\frac{1}{2}\sum_{k,l=1}^{n}(\alpha_{k}-\alpha_{k}^{*})(\alpha_{l}-\alpha_{l}^{*})\langle \Phi (s_{k}),\Phi (s_{l})\rangle \\ +\varepsilon \sum_{k=1}^{n}\left(\alpha_{k}+\alpha_{k}^{*}\right)-\sum_{k=1}^{n}y_{k}(\alpha_{k}^{*}-\alpha_{k})\}\\ s.t.\left\{ \begin{array}{l} \sum_{k=1}^{n}\left(\alpha_{k}-\alpha_{k}^{*}\right)=0\\ 0\leq \alpha_{k},\alpha_{k}^{*}\leq Ck=1,2,\ldots ,n \end{array}\right. \end{array}$$


Linear expressions in high dimensional space can be determined in order to solve the target position coordinates:
(17)$$\begin{array}{l} f(s)=\sum_{k=1}^{n}\left(\alpha_{k}-\alpha_{k}^{*}\right)\Phi (s_{k})\cdot \Phi (s)+b\\ =\sum_{k=1}^{n}\left(\alpha_{k}-\alpha_{k}^{*}\right)K(s_{k},s)+b \end{array}$$


In Formula (17), the kernel function $K\left(s_{k},s\right)=\Phi \left(s_{k}\right)\cdot \Phi \left(s\right)$ is introduced which is the inner product of $\Phi \left(s_{k}\right),\Phi \left(s\right)$. The kernel function can map the linear inseparable data in the input space (low-dimensional) into the linear separable data in the high-dimensional feature space (regenerated kernel Hilbert space) through a feature map. There are already many kinds of kernel functions. This paper adopted a Gaussian kernel function, which has strong anti-interference ability against noises or outliers in samples. The Gaussian kernel function is represented as:
(18)$$K(s_{k},s)=\exp\left(-\frac{{\left\|s_{k}-s\right\|}^{2}}{2\sigma^{2}}\right)$$
where σ is a user-specified parameter which indicates the width of the Gaussian kernel and ${\left\|\cdot \right\|}^{2}$ denotes the L^2^ distance between two input RSSI vectors.

The regression model of the vector and position information can thus be established through the constant optimization of parameters in the offline stage. In the online stage, the coordinates of the real-time position can be obtained by entering the collected real-time fingerprint data into the trained regression model.

It is thus evident that three parameters—insensitive coefficient *ε*, penalty factor *C* and width of Gaussian kernel σ—are critical during the regression process.

#### 3.3.2. The PSO Algorithm

The PSO algorithm is one of the most widely used optimization algorithms. It is characterized by simulating the cluster intelligence of birds and realizes global searching through cooperation and competition among individuals. It is widely used in neural network training, SVM parameter optimization and objective function optimization [36].

The mathematical description of the PSO algorithm is: let in one-dimensional space, a swarm consisting of *m* particles $X={\left(x_{1},x_{2},\cdots ,x_{m}\right)}^{T}$, the *q*^th^ particle position is $x^{q}={\left(x_{1}^{q},x_{2}^{q},\cdots ,x_{m}^{q}\right)}^{T}$ and its velocity is $v^{q}={\left(v_{1}^{q},v_{2}^{q},\cdots ,v_{m}^{q}\right)}^{T}$. The individual extreme value is $p^{q}={\left(p_{1}^{q},p_{2}^{q},\cdots ,p_{m}^{q}\right)}^{T}$ and the global extreme value of whole swarm is $p^{g}={\left(p_{1}^{g},p_{2}^{g},\cdots ,p_{m}^{g}\right)}^{T}$. According to the principle of following the optimal particle, particle $x_{i}$ updates its position and velocity as follows:
(19)$$v_{l}^{q}(k+1)=v_{l}^{q}(k)+c_{1}\times rand_{1}()\times (p_{l}^{q}(k)-x_{l}^{q}(k))+c_{2}\times rand_{2}()\times (p_{l}^{g}(k)-x_{l}^{g}(k)),$$
(20)$$x_{l}^{q}(k+1)=x_{l}^{q}(k)+v_{l}^{q}(k+1)$$
where $c_{1}$ and $c_{2}$ are the learning factor or acceleration constant and $rand_{1}(),rand_{2}()\in (0,1)$ is a random number. The *l*^th^ dimension position and velocity of the particle q in the *k^th^* iteration are $x_{l}^{q}(k)$ and $v_{l}^{q}(k)$ respectively. The individual extreme position of the particle q in the *l*^th^ dimension is $p_{l}^{i}(k)$. The global extreme position of the swarm in dimension *l* is $p_{l}^{g}(k)$.

The main steps in PSO are shown in Table 1:

#### 3.3.3. The SVR Optimized by PSO

Since the SVR performance depends on its kernel function width σ, penalty factor C and insensitive coefficient ε, the selection of these three parameters directly affects the learning and generalizability of SVR. The search rules for PSO are simple and easy to implement and the convergence speed is also fast. Based on these advantages, the PSO was adopted to optimize the parameters of the SVR model in this paper. The optimization process is as follows:
**Step** **1:**Determining the fingerprint training sample as the input of this algorithm;**Step** **2:**Using the PSO algorithm to determine the SVR optimal parameters:
**Step** **2.1:**Initializing the swarm: the population size is m, the *q*^th^ particle $p^{q}=(C_{q},\varepsilon_{q},\delta_{q})$ and its dimension n=3, so that the particle dimension equals 3. A Gaussian kernel function was adopted in this paper;**Step** **2.2:**Calculating the fitness value of each particle, expressed as:
(21)$$\left\{ \begin{array}{l} fit(s_{q})=\frac{1}{RMSE}\\ RMSE=\frac{1}{h}\sum_{k=1}^{h}\frac{{\left(f(s_{k})-y_{k}\right)}^{2}}{n} \end{array}\right.$$
where $f(s_{k})$ is the predicted value of the *k*^th^ fingerprint sample tested; $y_{k}$ is the actual value of the *k*^th^ test sample; h is the number of test samples;**Step** **2.3:**In each iteration process, updating the individual extremum $p_{d}^{k}$ and global extremum $p_{d}^{g}$ of each particle according to the fitness value of each particle;**Step** **2.4:**Updating the velocity and position of particles according to Formula (19) and (20);**Step** **2.5:**If the fitness value has met some requirements or the maximum iteration number has been reached, the algorithm stops and global extreme value is output; otherwise, return to Step 2.1.
**Step** **3:**Take the optimal value determined by the algorithm as the optimal parameter of the model.


The flowchart of the PSO-SVR algorithm is shown in Figure 3.

## 4. Verification in a Real-Life Application 

The proposed algorithm is applied to a dangerous area monitoring system in a chemical factory area. In order to verify its performance, the traditional positioning algorithm and the positioning algorithm proposed in this paper are experimentally analyzed by using the fingerprint data collected in the real environment.

### 4.1. Verification Environment and Standards 

#### 4.1.1. Factory Dangerous Area Monitoring System

Building a smart factory in a coking company faces many challenges, including complicated processes, a variety of production equipment, an extensive production workshop and numerous hazardous sources such as sulfuric acid. A chemical workshop of this coking company as shown in Figure 4a was selected for this study. By deploying a new WiFi system, the company designed a high-precision indoor positioning system to monitor dangerous areas and assist in emergency evacuation. 

The application scenario is shown in Figure 4b. An array of wireless routers was arranged above the dangerous area. Each worker was equipped with a specially designed mobile device that contained a WiFi module ESP8266 and thus could read RSSI signals, which is showed in Figure 5. Please note that this mobile device is being redesigned and will be eventually embedded in a badge ID card. In the offline stage, all fingerprint data were sent to a server to be clustered and used to train an SVR prediction model. Then this trained model was downloaded to the mobile devices. In the online stage, the WiFi circuit in the mobile device was receiving RSSI signals, and always localizing itself according to the position prediction model which was running totally in the mobile device. When a person entered the dangerous area, an alarm would be triggered in his/her mobile device and the actual position information would be sent to the server. If the mobile device was not authorized, an alert would be issued.

#### 4.1.2. Indoor Environment

The floor plan of this experimental chemical workshop is depicted in Figure 6. It covered an area of approximately 677.22 m^2^, of which the length was 31.75 m and the width 21.33 m. A rectangular coordinate system was established within the northwest corner of the workshop as the origin of coordinates. One hundred and sixty RPs (blue points in Figure 6 were evenly deployed in the room, with an interval of two meters. There were 11 APs, and the sampling interval was set to one second, thus obtaining a total of 9600 samples within the acquisition time of 60 s. For simulating the actual scenario and comparing different algorithms, a wireless scanning tool inSSIDer and laptop ThinkPad L440 with an intel Core i5-4300M (2.6 GHz/L3 3 M) processor and a wireless network card were used to collect fingerprints and other information such as physical addresses, corresponding MAC addresses etc. The preprocessed information formed a fingerprint sample after relevant information was extracted. Each location fingerprint sample included its position coordinates and RSSI values collected at this point. The processed fingerprint data were stored in the database for further task and its format is shown in Table 2.

#### 4.1.3. Evaluation Criteria

The positioning performance of the indoor positioning system was mainly evaluated from the aspects of accuracy and precision. This paper evaluated the advantages and disadvantages of the positioning algorithm from two factors: the average positioning error and the cumulative error distribution function.
(1)Mean absolute error (MAE):MAE is the most commonly used indicator to measure positioning precision. The larger the value, the lower the accuracy of the positioning algorithm:
(22)$$MAE=\frac{1}{N_{t}}\sum_{i=1}^{N_{t}}\sqrt{{\left(\hat{p}(i)-p(i)\right)}^{T}\left(\hat{p}(i)-p(i)\right)}$$
where, $\hat{p}(i)$ and p(i) are the estimated and actual position of the *i*^th^ tested point respectively.(2)Cumulative distribution function (CDF):The positioning distance error distribution was measured by CDF, which can intuitively reflect the percentage of positioning times in which the error is under a certain threshold:
(23)$$F(e)=P(e_{i}\leq e)=\frac{N(e_{i}\leq e)}{N}$$
where $N(e_{i}\leq e)$ is the number of tested points $e_{i}\leq e$ and *N* is the total number tested points for positioning. By comparing the accuracy of the different positions of the algorithm, it can be seen that the higher value of the CDF under a certain position error threshold, the higher the positioning accuracy and the better the effect.


### 4.2. Verification of the Influence of Fingerprint Density on Positioning Accuracy

Since the fingerprint density has an important influence on the positioning accuracy, a special experiment was designed to verify this effect. Fifty additional RPs were uniformly distributed in the target location area (Figure 6) and used to construct a larger fingerprint database together with the original 160 RPs. These 50 RP fingerprints could be obtained either by measuring actually or by the GPR method mentioned above. It was verified that the difference between the RP fingerprints with these two methods is very small. The algorithm KNN (k = 5) was used to compare the positioning accuracies before and after 50 RPs were added. As shown in Table 3 and Figure 7, it can be seen that the average absolute error of the positioning was 2.954 m and 2.364 m respectively. Although the average positioning error of both scenarios were above 2 m, the positioning error decreased 19.97% given a denser fingerprint database. The CDF curve of position errors in the denser fingerprint scenario was always on the left of that with original fingerprint database. In the case of positioning error less than 2 m, the probability in the denser fingerprint scenario was 53%, and it was about 33% in the original fingerprint scenario. All of the results mean that the denser the RPs, the higher the positioning accuracy.

### 4.3. Clustering Algorithm Analysis

#### 4.3.1. The Effect of Clustering Technology on Location Performance

In this paper, the APC algorithm was firstly used to conduct partitioning in RP to narrow the positioning area; PSO-SVR was then used to achieve accurate positioning. The positioning algorithm APC-PSO-SVR is compared with the algorithm without clustering PSO-SVR in Table 4 and Figure 8. Table 4 indicates that the localization accuracy of APC-PSO-SVR was, on average, 1.478, while that of PSO-SVR was 2.174. The APC-PSO-SVR improved the localization accuracy over 30%. Thus, the clustering in the coarse localization step played an important role in localization accuracy. As it can be seen from Figure 8, the APC-PSO-SVR curve was on the left side of the PSO-SVR curve, indicating that, if the same localization error (*x*-axis value) was set in APC-PSO-SVR and PSO-SVR, the ratio of the former was higher (*y*-axis value), meaning that the overall localization accuracy of APC-PSO-SVR was thus higher.

#### 4.3.2. Effect of Similarity Selection on Positioning Performance

To further evaluate the influence of different fingerprint similarity calculation methods on localization performance, the Euclidean distance similarity, the cosine similarity and the Shepard similarity discussed in Section 3.2.1 were respectively used as the similarity metric, and the localization performance was compared (Table 5, Figure 9).

The average position errors of the indoor localization algorithm based on clustering improvement with Euclidean distance, cosine similarity and Shepard similarity were 1.709 m, 1.503 m and 1.478 m. In Figure 9, the CDF curve from Shepard similarity was on the left side of the curve from the two other similarity metrics. The experimental results showed that the Shepard similarity could better measure the similarity between real-time and stored fingerprints and could improve the localization accuracy compared with the other two similarity metrics.

### 4.4. Analysis of Algorithm in Fine Positioning Phase

In order to analyze the effectiveness of the positioning method proposed in this paper, the performance of system using KNN and back-propagation neural network (BPNN) was compared with that of the system using PSO-SVR in the fine localization stage. The experimental results are shown in Table 6 and Figure 10.

As it can be seen from Table 6, the mean localization accuracy of PSO-SVR was improved by 22.4% and 29.3% respectively than that of KNN and BPNN. In Table 6, when the error threshold was set to 50%, the position errors of PSO-SVR, BPNN and KNN were 1.28 m, 1.732 m and 1.755 m respectively, which also shows that the algorithm of PSO-SVR had better performance than the other two algorithms. In Figure 10, the curve of CDF based on PSO-SVR was on the left side of the curve based on the other two algorithms as well as that shown in Figure 8 and Figure 9. Experimental results of Figure 10 demonstrate that performance based on BPNN and KNN was nearly similar and, compared to BPNN and KNN, the localization algorithm PSO-SVR could significantly reduce the localization error and improve the localization accuracy of the system.

### 4.5. Performance Analysis

#### 4.5.1. Algorithm Complexity and Storage Requirements 

In Reference [34], the algorithm complexity was compared in detail among the SVR algorithm, the ANN algorithm and the probability-based algorithm. It concluded that the SVR algorithm has the advantage of minimal complexity and storage requirements, while the other two algorithms are so complicated and require such large storage that they cannot run in a mobile device. Thus, when transferred to services, they will aggravate the communication loads, reduce the battery life and slow down the localization process. The algorithm complexity of the proposed algorithm in the offline phase was a little higher than the SVR algorithm mentioned above because of the loop computations in PSO. However, because it could be placed in the background server, and its total number of computations was not large, the PSO-SVR algorithm in the offline phase was not the bottleneck of the system.

The advantage of two-level positioning was to reduce the computational complexity and storage requirements at the online stage. The computational complexity can be represented with Formula (24) given in Reference [9]:
(24)$$N_{sim}=N_{c}+N_{i}$$
where $N_{c}$ is the number of RP clusters and $N_{i}$ is the number of RPs in the cluster. In our scenario, the RPs were often clustered into six to twelve classes, and the maximal number of RPs in a cluster was around 40, so $N_{sim}$ was almost a constant which is around 52. In the actual operation, the complexity can be further reduced since only a coarse positioning would be run to compare the fingerprint between the cluster center and the mobile terminal. If the value was less than some threshold, the further fine positioning will not be carried out.

In Reference [28], the response time results of a two-level localization algorithm similar to ours are reported; some online prediction algorithms, such as Kernel based method, SVR and ANN are compared. In normal conditions, their running time and the difference among them are very little, which are less than 1 ms. It is enough for a system to respond in real time. Therefore in a two-level localization algorithm, the computation complexity of the online fine positioning algorithm no longer has a big impact on the overall system performance.

In addition, the storage requirement and communication load of this algorithm was very small. In contrast to some algorithms, which compare the fingerprint between the tested point and all RPs, the proposed algorithm only stores the prediction model, and its size was less than 100 K bytes, far smaller than the entire fingerprint database (about 6 M bytes in our application). At the same time, only some warning and request messages were sporadically transmitted between the mobile terminal and the server.

#### 4.5.2. Localization Accuracy

In Table 4 and Figure 8, the positioning error of the algorithm was 1.478 m on average; and 1.28 m under the accuracy threshold at 50%. In Reference [35], some popular algorithms for indoor fingerprint localization are listed. It can be noted that the position error was related to the RP interval and that the error was mostly 1.5 m in the case of 2 m RP interval and 50% accuracy. In Reference [34], some localization algorithms were reviewed and a similar result can be observed, and most position errors with these algorithms are 1~2 m. Even though there is some difference in verification circumstances, it can be concluded that the proposed algorithm was advantageous in accuracy.

In our application scenario, the worker uses a self-made mobile device containing a WiFi module to measure RSSI and localization, in which the battery power was limited. From the perspective of battery life and positioning accuracy, the SVR-based and two-level positioning method was the most suitable for monitoring dangerous areas.

## 5. Conclusions and Future Work

One of the core technologies for disaster management is the indoor positioning system, which can help rescue more people and prevent property loss due to the accurate and rapid localization of people and assets at the beginning of a disaster.

However, due to the fluctuation of wireless signals, the change in environments and monitoring areas, and the increase of reference points, indoor fingerprint positioning systems often face the contradiction of positioning accuracy, computational complexity and storage requirements. This paper proposed a two-level positioning method to address the challenge: in the coarse positioning stage, an APC algorithm based on Shepard similarity was used to realize clustering; then, in the fine positioning stage, an SVR with PSO parameter optimization was used for precise positioning. Both experiment results and actual implementations proved that the proposed method was more suitable than other methods in term of algorithm complexity, storage requirements and localization accuracy in dangerous area monitoring 

Future work will focus on the miniaturization of the mobile device and the integration of fingerprint indoor positioning with pedestrian dead reckoning (PDR) positioning. In fire scenarios, where the environment changes quickly, the fusion of indoor fingerprint positioning and PDR can be used for disaster management. Other sensors, such as magnetometers, gyroscopes and accelerometers can also be considered for integration with indoor RSSI fingerprint positioning to improve positioning performance.

## Figures and Tables

**Figure 1 sensors-19-04243-f001:**
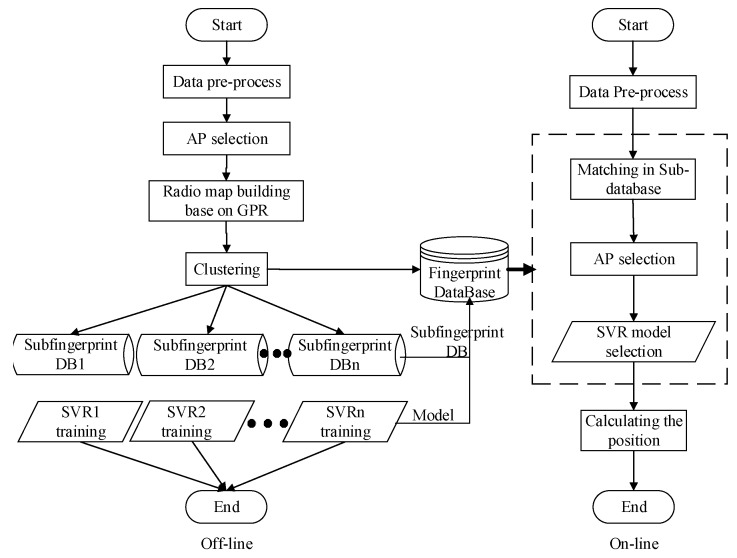
Overview of localization algorithm.

**Figure 2 sensors-19-04243-f002:**
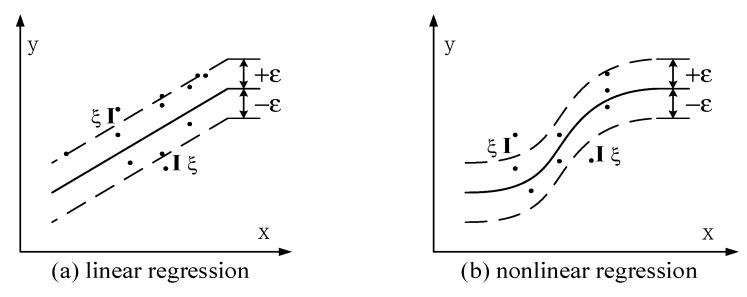
The overview of localization algorithm.

**Figure 3 sensors-19-04243-f003:**
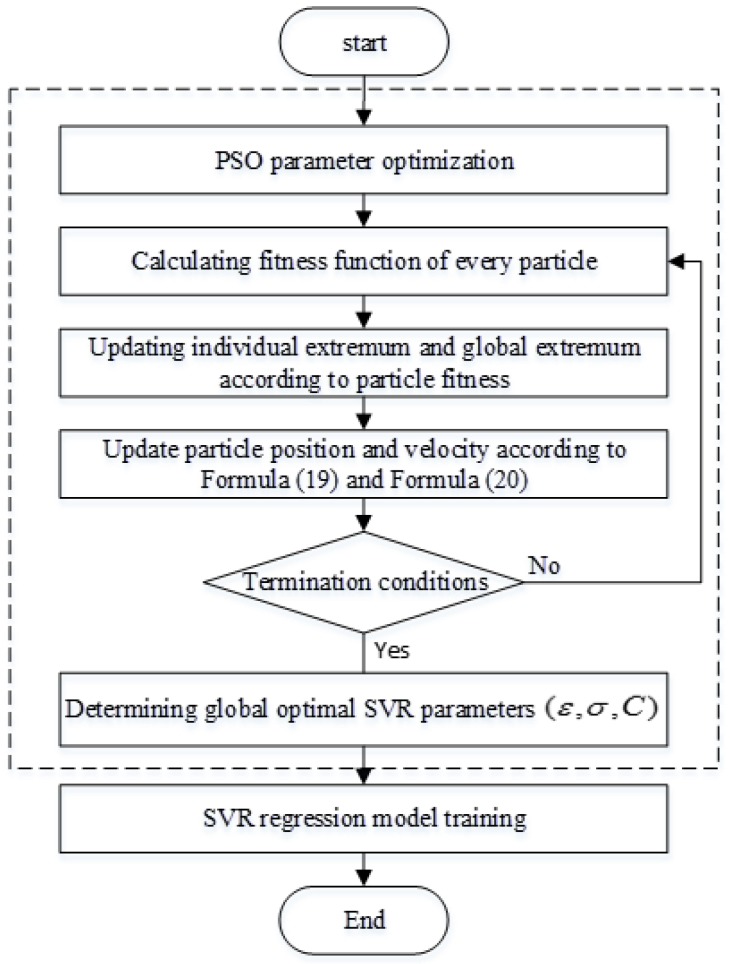
The flow chart of PSO-support vector regression (SVR).

**Figure 4 sensors-19-04243-f004:**
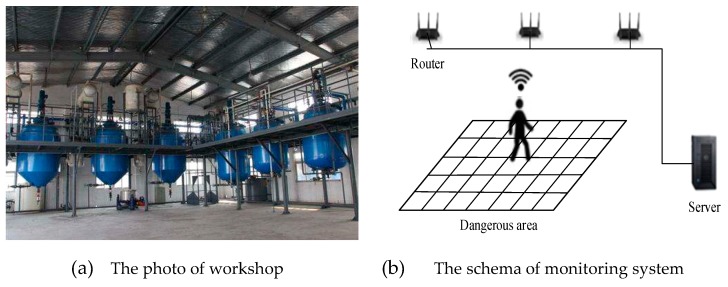
The chemical workshop and the schema of the positioning system.

**Figure 5 sensors-19-04243-f005:**
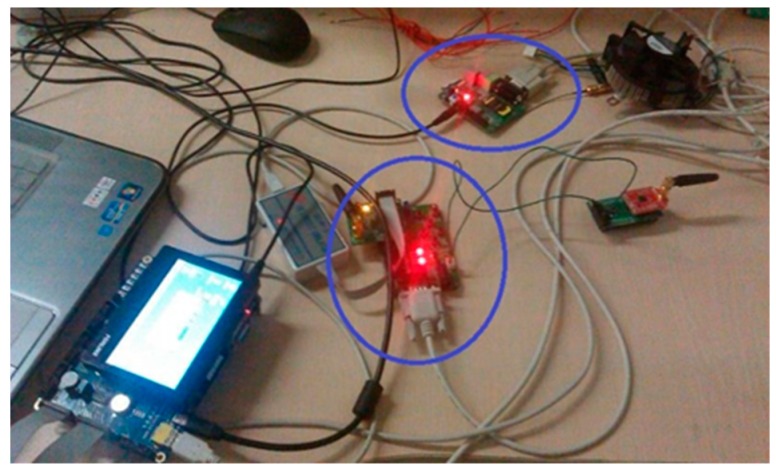
The specially designed mobile device.

**Figure 6 sensors-19-04243-f006:**
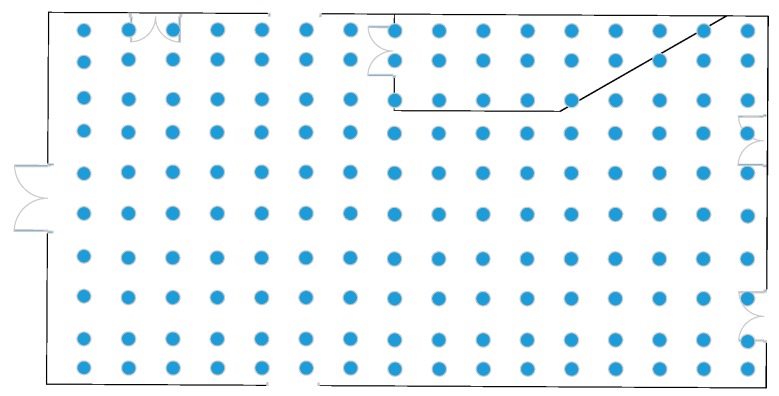
The chemical plant floor plan of a coking company.

**Figure 7 sensors-19-04243-f007:**
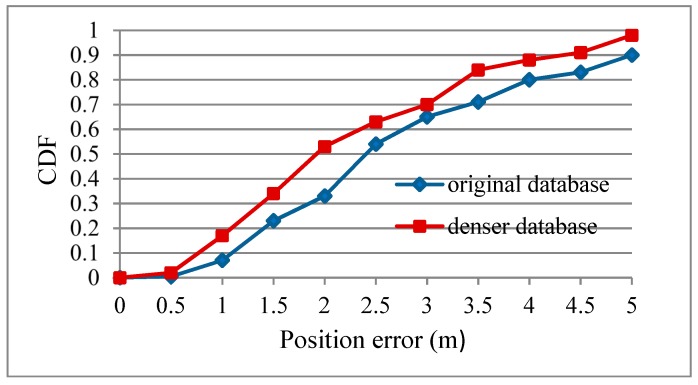
Cumulative distribution function (CDF) with different fingerprint databases.

**Figure 8 sensors-19-04243-f008:**
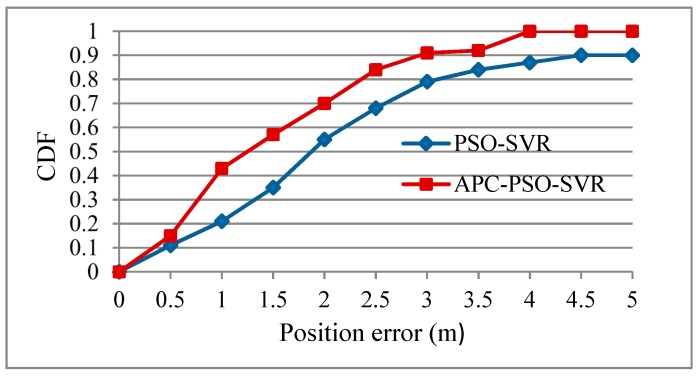
CDF before and after clustering.

**Figure 9 sensors-19-04243-f009:**
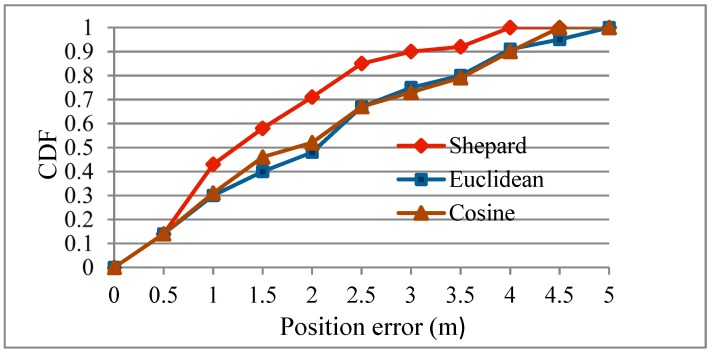
CDF with different similarity metrics.

**Figure 10 sensors-19-04243-f010:**
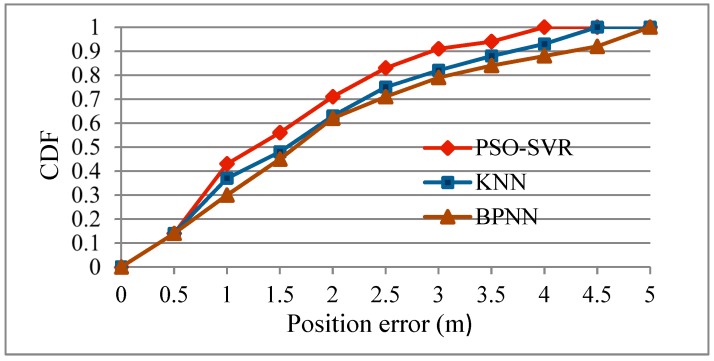
CDF with different localization algorithms.

**Table 1 sensors-19-04243-t001:** The particle swarm optimization (PSO) algorithm.

Step	Content
1	Initializing the swarm, including the population size M, the position $x^{q}$ and velocity $v^{q}$ of each particle.
2	Calculating the fitness value of all particles: *fit (*q*).*
3	Comparing the *fit (*q*)* in its current position with that in the best position it has passed through. If the new value is greater, replace the individual extreme.
4	Comparing the *fit (*q*)* in its current position with that in the best position all particles have passed through. If the new value is greater, take the current position as the global best position.
5	Updating the position and velocity with Formulae (19) and (20).
6	Repeating steps (2–5) until the stop condition is met or the iteration reaches the maximal number.
7	Obtaining every particle’s best position, local optimal solution, global position and global optimal solution.

**Table 2 sensors-19-04243-t002:** Fingerprint data.

RP coordination	AP1	AP2	AP3	AP4	AP5	•••
(1, 1)	−85	−65	−84	−91	−66	•••
(1, 2)	−83	−71	−80	−84	−72	•••
(2, 1)	−74	−69	−80	−83	−69	•••
(2, 3)	−82	−74	−82	−88	−73	•••
•••	•••	•••	•••	•••	•••	•••

**Table 3 sensors-19-04243-t003:** Comparison of positioning performances with different fingerprint databases.

Fingerprint Database	Mean Position Error (m)	Error under Threshold 50% (m)	Error under Threshold 90% (m)
The original database	2.954	2.275	4.915
The denser database	2.362	1.923	4.547

**Table 4 sensors-19-04243-t004:** Comparison of two positioning performances before and after clustering.

Algorithm	Mean Position Error (m)	Error under Threshold 50% (m)	Error under Threshold 90% (m)
PSO-SVR	2.174	1.961	4.473
APC-PSO-SVR	1.478	1.28	3.5

**Table 5 sensors-19-04243-t005:** Comparison of positioning performance using different similarities.

Metric	Mean Position Error (m)	Error under Threshold 50% (m)	Error under Threshold 90% (m)
Euclidean distance	1.709	2.275	3.847
Cosine	1.503	1.923	3.915
Shepard	1.478	1.28	3.5

**Table 6 sensors-19-04243-t006:** Comparison of positioning performance using different localization algorithms.

Algorithm	Mean Position Error (m)	Error under Threshold 50% (m)	Error under Threshold 90% (m)
KNN	2.091	1.755	4.092
BPNN	1.905	1.732	3.537
PSO-SVR	1.478	1.28	3.5

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
