# Peer review of "A Two-Level WiFi Fingerprint-Based Indoor Localization Method for Dangerous Area Monitoring"

_sensors, 2019, doi:10.3390/s19194243_

Round 1

Reviewer 1 Report

The contribution of this paper is limited. The proposed techniques are more-or-less straightforward. 

The writing needs a deep revision.

The considered techniques seem to have essentially the same performance (differences below 0.5m in 3-4m ranges).

Equation (19) do not make sense. Either it has a typo or the result is 0. 

The variable p has multiple meanings in the text. And it is not clear when it is a scalar or a vector (the vectors should be in bold, non-italic).

It is not clear what is the implementation complexity of the proposed technique and how it compares with other techniques from the literature. 

Reviewer 2 Report

The paper concerns a WiFi-based indoor positioning system that uses a "hierarchical" approach with coarse and fine localization stages, and applies the affinity propagation clustering to cluster the radio map into smaller chunks, and the support vector regression for final positioning. Parameters of the latter algorithm are found using the particle swarm optimization aproach.

The paper is reaonably well written, ans presents the algorithms in a way that is rather easy to follow, which is an advantage.  Also the comparison of methods in the experimental part is interesting.

However, there are some drawback that should be addressed before publication:

The motivation related to the disaster response use cases is quite weak. An analysis of such a scenario in the Introduction would be appreciated, e.g. how many people uses the system, do they use smartphones or other devices, how the solution scales to large areas and with the number of access points. Related work should address in a more comprehensive way the off-line map building. this is a tedious task, and the recent results related to crowdsourcing, multi-user systems and/or other techniques should be included. I think that the seminal paper R. Eberhart, J. Kennedy, “A new optimizer using particle swarm theory”. In: MHS’95. Proceedings of the Sixth International Symposium on Micro Machine and Human Science, 1995, 39–43 should be cited in the context of PSO.  The mathematics should be edited more carefully, avoiding such formatting as "1/p" in (1) - rather use LaTeX properly.  Also the figure with text in Chinese (Fig. 5) should be avoided in an international paper.  The quality of Fig. 6 is also questionable. Please, ask a native speaker to improve English.

Reviewer 3 Report

This paper proposed an indoor localization algorithm based on affinity propagation clustering (APC) algorithm and support vector regression (SVR) algorithm, to balance the accuracy and response time in indoor Wi-Fi localization after an accident or in the case of dangerous areas monitoring. It exploits APC to divide fingerprint database into several sub databases, thus the response time can be cut down. Meanwhile uses SVR with its parameters being optimized by particle swarm optimization (PSO) for fine positioning, improving the positioning accuracy.

However, some parts are not clearly described and difficult to be understood, it results in reading interruption and unconvincing conclusions.

The schemes of algorithms lack detailed information and thus, are hard to repeat. For instance:

1) In section 3 Proposed Indoor Localization Method, there is no detail about data pre-process and coarse localization algorithm.

2) Section 3.2 simply introduce the basic APC algorithm based on the shepard similarity metric, it is unclear how to use APC algorithm to divide origin fingerprint database into different fingerprint subsets in offline stage.

3)In addition, it is unclear how to determine which fingerprint subset of real-time RSS sequence belongs to and which SVR model should be used to find the fine position.

In experiment, some parts are not clearly described, and experiments are insufficient and the conclusion is not credible, such as.

1) Real experimental scene are not clearly described, and experimental setup and platform is not given in this paper.

2) As the author said that a dangerous area monitoring system has been realized in a chemical factory area, where is the system working on, a laptop, or a mobile phone? And what is a badge card? How can a badge card collect RSS fingerprint? Besides, as a badge card can collect RSS, why authors use the existed wireless scanning tool insider to collect RSS fingerprint, it is very confused.

3) The comparative experiment is insufficient to support the conclusion, as there is no other existed clustering algorithms are compared with APC during positioning process.

4) There is no experiment for verifying the response time of localization system is cut down.

Some conspicuous grammatical and format errors throughout the paper need to be corrected.

Round 2

Reviewer 1 Report

The authors improved substantially the paper.

Author Response

very thank you for your positive comments. It is your suggestion let me know how to improve the article. Thanks again

Reviewer 3 Report

Although the authors have made some modifications, the content of the modification is not clear. There are serious problems in the format of the paper, which cause great difficulties in reading.

In line 192-198: In theory, the more dense the mesh, the more precise the localization accuracy. No data to prove the statement. Need further experiment. Consider that the more dense the mesh, the more similar between adjacent grids.

In Figure1 and Figure 3, which figure is used, the first or the second? In line 183-191: What is the reason behind that authors choose the first criteria as the coarse localization method?  

In line 423-434: It’s not convincing as there is only a schematic diagram of positioning system, without figure of real objects of specialized badge card and testbed in real-world.

The comparative experiment is insufficient to support the conclusion, as there are many existed clustering algorithms are used to fingerprint database segmentation,  for instance, K Means, Fuzzy C Means (FCM), and Affinity Propagation have been well deployed in some positioning systems for data segmentation. How can authors say the APC clustering with good effect in this specific case without any experimental verification?

The format of the revised paper is so poor, authors should be refined it seriously and carefully.
